# The Quasi-History of Early Quantum Theory

Oliver Passon 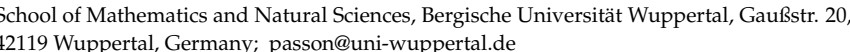

School of Mathematics and Natural Sciences, Bergische Universität Wuppertal, Gaußstr. 20,
42119 Wuppertal, Germany; passon@uni-wuppertal.de

**Abstract:** While physics has a rather ahistoric teaching tradition, it is common to include at least anecdotal reference to historical events and actors. These brief remarks on the history are typically distorted. I take issue with the textbook narrative of the historical development of early quantum theory and rectify some of the more severe misrepresentations. This seems to be all the more important, since the history of physics is commonly (and rightly) regarded as a means to foster scientific literacy and a more appropriate understanding of the nature of science (NoS).

**Keywords:** history of physics; Planck's law; light-quantum hypothesis; photo-effect; Bohr's atomic model; Compton effect; photon; nature of science (NoS)





## 1. Introduction

Although science teaching mainly aims at imparting current practices and theories, it is often framed historically. However, this "quasi-history" [1,2] often distorts the actual course of events and presents the history of science as a cumulative sequence which finally led to the acceptance of the current theories [3]. Thus, textbooks produce a sense of participation in a certain methodological and social tradition which may be fictitious [4]. In view of current debates in physics education, this may be characterized as a failure to provide an appropriate view on the nature of science (NoS).

This paper deals with the quasi-history of the discovery of energy quantization and the quantum of radiation, i.e., the "photon") as a case study (Section 2). As a by-product, my investigation helps to trace another problematic issue in the field which is disturbing beyond the question of historical accuracy. I mean the naïve line of tradition which is typically drawn from Einstein's tentative light-quantum hypothesis in 1905 to the current understanding of "photons" as objects described by quantum electrodynamics (QED). The decisive differences between the early light-quantum and the QED photon are often glossed over. As discussed in Section 2.5, this quasi-history of early quantum physics distorts both the history and the physics of the photon.

In closing (Section 3), I suggest a brief but more appropriate representation of the historical events and provide some context within the current NoS debate in physics education.

## 2. The Quasi-History of the Early Quantum Theory

This case study investigates the quasi-history of the discovery of energy quantization and the quantum of radiation (i.e., the "photon") in the early quantum theory as presented in physics textbooks. First, I sketch such a "typical" account which is a collage of elements frequently met (to substantiate my claim further, include references and quotations from selected textbooks; it should be emphasized that the discussion is not intended to denigrate these otherwise recommendable texts):

- Quantum mechanics arose in 1900 from the explanation of the black-body spectrum. To avoid the (ultra-violet) UV-catastrophe of the classical prediction, Max Planck introduced the quantum of energy, $\epsilon = h\nu$, where $h$ is Planck's constant and $\nu$ denotes the frequency.

- In 1905 Albert Einstein generalized Planck's idea to the radiation field and introduced his light-quantum (with energy $E = h\nu$) to explain the photoelectric effect—a phenomenon which cannot be explained with the classical wave-theory of light.
- Niels Bohr in his atomic model from 1913 combined ideas from both Planck and Einstein. He proposed stable orbits for the electrons with fixed energies $E_i$. Any transition between these states is accompanied by an emitted (or absorbed) photon which meets the Bohr frequency condition $\Delta E = E_i - E_j = h\nu$.
- Arthur H. Compton's discovery of the effect named after him (i.e., the shift in wavelength of scattered X-rays) in 1922 and his explanation in terms of scattered light-quanta (soon to be called "photons") in 1923 convinced the last skeptics of Einstein's light-quantum hypothesis.

In showing how these steps oversimplify the actual course of events and contain even several plain errors, I can build on extensive scholarly work in the history of physics. Different aspects of this history have been closely investigated by [5–8] to mention just a few. More comprehensive overviews on the history of quantum theory have been given by, e.g., Jammer [9] and Mehra and Rechenberg [10].

Before starting, I would like to emphasize that while I try to correct several distortions and errors in the common quasi-history, I certainly do not intend to provide the "true" history of the events. The description surely fails to do full justice to the original motivation of the protagonists. The social dimension and will also oversimplify the development. In short, that this discussion also represents a specific narrative and some sort of "presentism", "Whiggism" [11], or "anachronism" [12] is both inevitable and maybe even desirable. However, the common quasi-history of quantum physics paints a naïve picture of a cumulative process in which anomalies ("UV-catastrophe"), successful predictions (photo-effect), or crucial experiments (Compton-scattering) drive the development. It is such a picture which is at odds with current views on the Nature of Science, i.e., my goal is to provide the necessary amendments to correct this *specific* misrepresentation.

Sections 2.1–2.4 deal with the abovementioned steps in the quasi-historical narrative.

### 2.1. Planck and the Black-Body Radiation Problem

Already in 1859, Gustav Kirchhoff argued that a black-body (i.e., a body that absorbs all incident radiation) in thermal equilibrium should emit a radiation spectrum which does not depend on its shape or material. Thus, there should be a universal function, $u = u(T, \nu)$, describing its spectral energy density at temperature $T$ in the frequency interval $[\nu, \nu + d\nu]$—an ideal testing ground for theoretical models. In 1896, Willy Wien suggested a radiation law from thermodynamic arguments [13]:

$$u_{\text{Wien}} = \frac{8\pi\nu^2}{c^3} \cdot \frac{\alpha\nu}{\exp{(\beta\nu/T)}}, \tag{1}$$

where $c$ denotes the speed of light. With the free parameters $\alpha$ and $\beta$ adjusted accordingly, this expression could describe the available data well. Discrepancies showed up in the late summer of 1900.

Another specific theoretical prediction for the black-body spectrum is the so-called Rayleigh-Jeans (R–J) law, which follows from applying the equipartition theorem to the continuous Maxwell radiation field:

$$u_{\text{R–J}} = \frac{8\pi\nu^2}{c^3} \cdot kT, \tag{2}$$

where $k$ is the Boltzmann constant.

However, the R–J law cannot be completely valid since it diverges in the high-frequency regime ("UV catastrophe"). Now, many textbooks (e.g., by Tipler and Llewellyn [14] (p. 122f))

suggest that the failure of this "classical" prediction was the great puzzle which initiated Planck's discovery of his radiation law in 1900 [15] (see [16] for English translation):

$$u_{\text{Planck}} = \frac{8\pi\nu^2}{c^3} \cdot \frac{h\nu}{\exp\left(\frac{h\nu}{kT}\right) - 1},\qquad(3)$$

including the discontinuous energy element, $\epsilon = h\nu$. This claim is incorrect for various reasons—the simplest being that the R–J law was only published in 1905, i.e., five years after Planck's law [17]. It is true, however, that Lord Rayleigh had anticipated this law already in 1900 in a brief note but introduced an exponential damping factor to avoid the divergence [18]. Planck did not cite Rayleigh's paper in his publications on the radiation problem from 1900 or 1901. However, given that Rayleigh's paper was cited by the experimenters (e.g., Rubens and Kurlbaum [19]), Planck must have known it. In any event, Planck's law displayed the $u \propto T$ relation for large wavelength which was found experimentally and also suggested by Rayleigh's argument.

Rayleigh returned to this issue only in 1905 and also provided the numerical factors which were missing in his publication from 1900. However, he committed a small mistake, and his result was too big by a factor of eight. James Jeans corrected this mistake immediately—thus the R–J law got its double name. The role of Jeans is discussed by McCaughan [20] who even suggested one possible origin of the quasi-historical narrative: It was Jeans himself who reported for the Physical Society of London on the history of the radiation problem in 1914. With respect to the R–J law, he stated: "This formula was given by Lord Rayleigh and the present author in 1900 [...]" [21]. This wrong date also entered the (much extended) second edition of the report in 1924 and made a textbook career thereafter. Another curious mistake can be found in the textbook of Alexander Komech. He claims without reference that the R–J law was already published in 1894 [22] (p. 12). Now, Jeans was a gifted student but entered Trinity College in Cambridge only at the age of 19 in October 1896. The common claim (see, e.g., [22], p. 15) that Planck in 1900 interpolated the radiation laws of Wien and R–J is obviously untenable for the same reason). To call the R–J law the "classical" prediction is also misleading, since the application of the equipartition theorem was debated at that time. For example, the failure of the rule of Dulong–Petit for the specific heat compromised this theorem, and it is well documented that it played no role in Planck's paper [5]; the term "catastrophe in the ultra-violet" was coined by Ehrenfest only in 1911.

In addition, it is unclear whether Planck in 1900 intended any *physical* energy quantization at all, a debate initiated by Thomas S. Kuhn in 1978 [6]. In Passon and Grebe-Ellis [23], a reconstruction of the different strands in this complex debate is provided.

Summing up, according to current historiography, Planck's study was no reaction to an anomaly, and it is doubtful whether Planck intended any quantization at all. However, be that as it may, nobody in the community picked up on "quantization" for many years anyway. Helge Kragh notes [24] (p. 63):

> "If a revolution occurred in physics in December 1900, nobody seemed to notice it, least of all Planck. During the first five years of the century, there was almost complete silence about the quantum hypothesis, which somewhat obscurely was involved in Planck's derivation of the black-body radiation law. The law itself, on the other hand, was quickly adopted because of its convincing agreement with experiment."

This is not the reaction one would expect if the very foundation of "classical physics" has just been shattered. Part of the explanation lies in Planck's rather obscure derivation of his law (it is evidently this obscurity, which fuels the controversial debate on its interpretation, I indicate above). In addition, one observes how claims of validity are negotiated within the scientific community. The problem of black-body radiation was a rather specific one, and the finer details of this study were of little concern to many. Kragh [24] (p. 69f) points out that Einstein's later application of Planck's distribution law to the problem of

specific heat in 1907 (later extended by the Dutch theoretician Peter Debye [25]) played an important role since this was a more traditional field of physics. However, earlier than that Einstein introduced his famous light-quantum hypothesis; the second step in my reconstruction of the quasi-history.

### 2.2. Einstein and the Light-Quantum Hypothesis

In 1905, Einstein published several remarkable papers. Among them is the famous "Concerning an heuristic point of view toward the emission and transformation of light" [26] (see English translation in [27]). The usual textbook account caricatures its content as follows [28] (p. 12f):

> "The hypothesis of the existence of the quantum of light was introduced by Albert Einstein in 1905 starting from Planck's solution to the black-body problem. In this way, he was able to explain the photoelectric effect, i.e., the emission of electrons by a metal surface when it is illuminated by light."

or [14] (p. 129):

> "Einstein assumed that the energy quantization used by Planck in solving the black-body radiation problem was, in fact, a universal characteristic of light. Rather than being distributed evenly in the space through which it propagated, light energy consisted of discrete quanta, each of energy $hf$."

Historians of physics are quick to point out that almost every statement in these accounts is imprecise or wrong; Einstein's reference to Planck's radiation law is only marginal, the photoelectric effect is only one application mentioned (not even very prominently), and it is not "explained" but predicted, given that conclusive data became available only in 1914. Let us briefly set the record straight.

Einstein motivated his investigation with the "profound formal difference between the theoretical conceptions physicists have formed about [...] ponderable bodies, and Maxwell's theory of electromagnetic processes [...]". While material bodies are thought to consist of a "finite number of atoms and electrons" the electromagnetic field is described by continuous spatial functions [27] (p. 86). Einstein suggested that this asymmetry might produce problems in situations where material bodies and radiation interact, e.g., in the case of black-body radiation. With hindsight, it appears that Einstein anticipates a full-blown "quantization program" here in which fields and matter are treated on the same footing. While this assessment would mean to recast past developments in terms of current theories, Einstein's remarks are surely long-sighted. In any event, the paper is so good that its presentation needs no distortion.

The main body of the paper began by treating exactly this black-body radiation problem. Here, Einstein derived (independently) the "Rayleigh-Jeans law" (as it is now called) and emphasized the divergence of the total energy. Since this derivation was based on Maxwell's theory and (statistical) mechanics, it illustrates the problems caused by the asymmetry mentioned above. Einstein noted further that Planck's law approximates this expression in the large wavelength regime which allowed for the determination of the Avogadro constant. In the remainder of the paper, Einstein was silent on Planck's law.

The mathematical core which led to the introduction of light-quanta was based solely on Wien's radiation law (Equation (1)) from 1896. Einstein derived the entropy $S_0$ of the radiation in the frequency interval $[\nu, \nu + d\nu]$ contained in a volume $V_0$. A similar expression $S$ holds for the entropy of a sub-volume $V$. From the entropy difference $S - S_0$, Einstein calculated the probability of an energy fluctuation into the sub-volume $V$:

$$W_{\text{rad}} = (V/V_0)^{\frac{N}{R}\frac{E}{\beta\nu}}. \tag{4}$$

Here $N$ denotes the Avogadro constant and $R$ the gas constant. Einstein then calculated the corresponding probability of $n$ independent particles fluctuating into a sub-volume:

$$W_{\text{gas}} = (V/V_0)^n. \tag{5}$$

Comparing these equations, Einstein suggested that [27] (p. 97)

"[...] monochromatic radiation of low density (within the range of validity of Wien's radiation formula) behaves thermodynamically as if it consisted of mutually independent energy quanta of magnitude $R\beta\nu/N$."

The wording was chosen carefully, i.e., there was no claim about, say, light in general consists of quanta, since the analysis dealt with a specific frequency regime only. However, Einstein introduced light-quanta heuristically as localized and distinguishable ("independent") entities.

Note also, that not even Planck's constant $h$ was used in that paper. However, with $\beta = h/k$ and $k = R/N$, the exponent of Equation (4) turns into the familiar expression $E/h\nu$.

Einstein closed the paper with short sections on possible applications. These were (i) the Stokes rule of photo-luminescence, (ii) ionization of gases by ultra-violet light, and (iii) the photoelectric effect. Here the famous "Einstein equation" (in modern notation) $T_{\max} = h\nu - W$ was introduced, with $T_{\max}$ the kinetic energy of the photo-electrons and $W$ the work function (I note in passing,, that the actual measurements (e.g., Millikan's stopping potential method) are not sensitive to the work function of the cathode, i.e., the material that emits the electrons. Due to contact potentials, one surprisingly measures rather the work function of the anode [29]).

While Millikan could confirm this Einstein equation in 1914 [30], this result did not even convince himself of the validity of the light-quantum hypothesis. Apparently most physicists joined him in this assessment because there were also competing models to account for the data [7]. In addition, Millikan's experiment measured only the maximal energy of the photo-electrons and not the energy spectra. It was the paper by Maurice de Broglie in 1921 which removed this ambiguity and had central importance for the growing acceptance of the light-quantum hypothesis [31] (p. 266ff).

Summing up, the light-quantum hypothesis was no application or generalisation of Planck's paper on black-body radiation . The explanation/prediction of the photo-effect was no main intention of Einstein and the confirmation of the "Einstein equation" only played a minor role in the gradual acceptance of the light-quantum hypothesis. Not least important, the frequent claim that this effect has no explanation in terms of the wave-theory of radiation is likewise incorrect. In fact, the photoelectric effect can be described by the semi-classical approximation in which the radiation is treated classically (see Section 2.5). Importantly, this semi-classical explanation also gets the angular distribution of the photo-electrons right, while the naïve corpuscular explanation fails in this respect [32].

*2.3. Bohr and the Atomic Model*

Bohr's atomic model of 1913 is discussed by many textbook authors as an *application* of Einstein's light-quantum hypothesis. Douglas Giancoli provides a typical example [33] (p. 789):

"In this Bohr model, light is emitted only when an electron jumps from a higher (upper) stationary state to another of lower energy. When such a transition occurs, a single photon of light is emitted."

A similar statement can be found, e.g., in [34], p. 7, or in [35], p. 76.

It is certainly true that Bohr's model puts electrons on stationary states with discrete energies $E_i$ and that the transition between these states is accompanied by "homogeneous" (i.e., monochromatic) radiation of the frequency $\nu = \Delta E/h$ [36]. However, Bohr took this to be just classical electromagnetic radiation which happens to have a specific frequency, and his rejection of the light-quantum hypothesis until the mid 1920s is well documented [10] (pp. 532–554). In his trilogy from 1913, Bohr stated explicitly that he tried to offer a solution which remained consistent "with experiments on phenomena for which a satisfactory explanation has been given by the classical dynamics and the wave theory of light" [36] (p. 19). This misrepresentation of Bohr's study was also noted by Stachel [37].

To view the Bohr model as a successful application of the light-quantum hypothesis should confuse textbook authors and readers anyway. It is well known (and often remarked)

that the light-quantum hypothesis was mostly rejected (until, say, the early 1920s), while the Bohr model was well received from the outset, could celebrate early successes, but passed its heydays in the early 1920s [38]. This would be rather odd (or even self-contradictory) if the latter would contain the former.

There is certainly a sense in which Bohr's model provided a radical departure from conventional electrodynamics since the radiation frequency is *different* from the orbital motion frequency. (Interestingly, Stachel [39] collects some evidence that Einstein may have, already anticipated Bohr's atomic model around 1905. In any event, he was very excited about its discovery. His derivation of Planck's radiation law from the study of transition processes [40] (see [41] for English translation) applied Bohr's idea of stationary states and provided another justification for the frequency condition $\Delta E = h\nu$—this time even for a non-periodic process.) However, as is well-known, the later developments (starting with Heisenberg's matrix mechanics) went in another direction and abandoned particle orbits and semiliteral models of the Bohr-type altogether. (As noted by Heilbron [42], the Bohr model was actually a late product of Victorian physics and stood in a line of work initiated by J. J. Thompson. Some of its success can be traced to the impact of the first world war. This great divide in modern Western history also interrupted research lines and precluded the development of alternatives. Note that much of the further developments of Bohr's model were conducted by Arnold Sommerfeld who escaped military service owing to his age. Heilbron [42] (p. 230) suggests that not only for general history is the first world war the watershed between the 19th and 20th centuries—but also for the history of physics).

Nicolaas P. Landsman remarks with respect to Bohr that "[h]is model probably would have gained in consistency by adopting the photon picture of radiation" [43] (p. 424). Thus, this Bohr episode provides a good example for the quasi-historical tendency to increase the coherence and to distort past events to fit them into a more rational narrative.

However, there is an additional twist to the Bohr story. When asked to contribute an article to the Einstein volume of the series *Living Philosophers*, Bohr famously wrote on his discussions with Einstein on epistemological problems. To set the stage, he briefly summarized the development of quantum theory including his atomic model from 1913. There one reads the following account of the radiation between stationary states [44] (p. 204):

> "[...] the spectra were emitted by a step-like process in which each transition is accompanied by the emission of a monochromatic light-quantum of an energy just equal to that of an Einstein photon."

This shortened account is clearly justified, given that Bohr aims at a different issue in this essay. While he speaks of a "light-quantum" he still does not call it "Einstein's light-quantum" but just quanta of the same energy as the "Einstein photon" (actually, Einstein did not use the term "photon" in any of his papers). Still, when considered superficially, this Bohr quotation apparently supports the quasi-historical account of the Bohr model. This nicely illustrates that quasi-historical accounts are also produced by the physicists involved in the development.

### 2.4. Compton and The Light-Quantum

Let us turn to the Compton effect, which allegedly played a central role in the final acceptance of the light-quantum hypothesis. For example, Tipler and Llewellyn [14] (p. 561) state:

> "Einstein's suggestion was not widely accepted until, over the next 20 years, Millikan's thorough experimental investigation of the photoelectric effect and Compton's discovery and explanation of the Compton effect provided incontrovertible evidence for the quantization of electromagnetic radiation, the field quantum being a particle we now call the photon."

In a similar vein is the following "historical remark" [28] (p. 35):

"We would like to end this Subsection with a historical remark. Einstein's interpretation of the photoelectric effect involving the corpuscular nature of light had not completely convinced the scientific community about the quantization of the electromagnetic field. In this respect, the Compton effect, where energy and momentum are conserved in each single collision, played the role of a definitive experimental evidence of radiation quantization and convinced even the most skeptical physicists."

Brush [45] has pointed out that it is difficult to provide clear evidence for such acceptance claims. The problem is that in scientific publications the authors rarely explain if and (especially) why they have changed their minds on certain issues. Apparently, such a claim is often based on anecdotal evidence by a few prominent physicists. Brush [45] (Ch. 7.9) suggests another way to study the impact of the Compton effect on the acceptance of the light-quantum hypothesis. It is analysed in terms of the reception it found in textbooks and reviews of that period, i.e., texts with a stronger commitment to give an extended description. It can be shown that starting arround 1926, reviews quoted the Compton effect more often as compelling evidence for light-quanta, while textbooks and popular articles of that time favored the photoelectric effect.

However, the above quotations call Millikan's and Compton's study "incontrovertible" or "definitive experimental evidence" for the "quantization of electromagnetic radiation, the field quantum being a particle we now call the photon". Evidently these authors insinuate that the current reader should be equally convinced and that this is not just a matter of historical contingency but scientific necessity. However, the situation is more complex here, and apparently the view on the relation between Compton effect and the light-quantum hypothesis changed again after 1925/1926.

With the advent of matrix and wave-mechanics, the so-called semi-classical approximation could be applied to known phenomena (according to Alexander Blum [46], this was not viewed as an approximation at first). Here, the continuous (i.e., unquantized) Maxwell-field is inserted into the Hamilton operator, i.e., matter is treated quantum mechanically and the radiation field classically. Since this method allows a description of the photoelectric effect and Compton scattering [47,48], there is apparently no need to introduce any field quantization at this point.

The power of the semi-classical treatment extends even to more than the simple kinematics of the Compton scattering. Compton's original analysis left the *intensity distributions* (in energy and angle) open. This gap was filled by the study by Oskar Klein and Yoshio Nishina in 1929 [49]. However, their derivation (taking full account of relativity and spin) still applied the semi-classical approximation for the radiation, even though at the time the beginnings of QED were already developing [50] (p. 233f). Against this background, the following textbook account is particularly disturbing [14] (p. 141, Note 16):

"It was Compton who suggested the name photon for the light-quantum. His discovery and explanation of the Compton effect earned him a share of the Nobel Prize in Physics in 1927."

Now, the invention of the name "photon" is usually attributed to the American chemist Gilbert N. Lewis [51]. Kragh [52] has discovered that this name was used before (and independently) by at least four different researchers. Priority belongs here to the American physicist and psychologist Leonard T. Troland, who already coined the expression in 1916 in his paper [53], published a year later. Note that to all of these scientists, "photon" did mean something different than Einstein's "quantum of light". Compton rightly deserves credit for the popularization of the term in its current meaning, and that was not a little thing: To call Einstein's light-quantum a "phot-on" on a par with "electr-on" or "prot-on" reflects the widespread recognition of this concept and, so to say, elevated it into into a higher rank. At the same time, the name "photon" reflects the tendency of reification of the underlying concept; cf. Section 2.5. However, more important is the second claim of the above quotation. In fact, it is not correct that Arthur Compton was awarded the Nobel prize for "discovery *and explanation*" of the Compton effect, but only for the "*discovery* of

the effect named after him" as Karl Siegbahn stated in his presentation speech at the award ceremony [54]. Siegbahn's speech contains another interesting remark which reveals why the distinction between "discovery" and "explanation" really matters:

> "[...] the Compton effect has, through the latest evolutions of the atomic theory, got rid of the original explanation based upon a corpuscular theory. The new wave mechanics, in fact, lead as a logical consequence to the mathematical basis of Compton's theory. Thus the effect has gained an acceptable connection with other observations in the sphere of radiation." [54]

Karl Siegbahn delivered this speech as a member of the Nobel committee. However—more relevant in this context—he was an eminent physicist himself who had received the 1924 Nobel prize in physics for his discoveries and research in the field of X-ray spectroscopy.

The above quote, however, means that the corpuscular underpinning of the effect was questioned—with reference to the newly discovered wave mechanics, i.e., Schrödinger's theory. Compton's not receiving the prize for the explanation of his effect was surely no oversight but pure intention (note that this is as with Einstein, who received the Nobel prize in 1922 for his discovery of the law of the photoelectric effect—and not for the proposed explanation, i.e., the light-quantum hypothesis).

Siegbahn's remark appears to be very much in the spirit of the times, as noted by Brown [50] (p. 228). Again, such a reception claim is hard to verify from research papers alone. An interesting source is provided by a paper written by Otto Halpern and Hans Thirring in 1929 [55]. This paper is contained in the publication series *Ergebnisse der Exakten Naturwissenschaften* (*Results of the Exact Sciences*), which was intended to provide a semi-technical overview of current developments for students and physicists, working in neighboring areas. Given that this two-part paper has roughly 200 pages, it could be almost called a textbook. After discussing the semi-classical treatment of the photoelectric and Compton effects according to Beck [47] and Schrödinger [48], Halpern and Thirring remark [55] (p. 444):

> "According to current understanding, there is no compelling reason to accept the existence of "light atoms". In how far their acceptance could be based on general considerations concerning the wave-particle parallelism shall not be treated here. This question reaches into the area of quantum electrodynamic, a subject matter only now in the process of arising." (translated by myself)

Hence, the confusion about the light-quantum has recurred in 1926/27 after its apparently being already settled in 1923 (surprisingly, this point is merely mentioned incidentally in the standard work on the Compton effect by Stuewer [56] (pp. 288–290)). The important point here is the following: the photon concept which eventually "reappeared" through the development of quantum electrodynamic has distinctly other features than Einstein's early conception or Compton's photon in his kinematic treatment of the photon-electron scattering. It is certainly no accident that it got the same name, but for conceptual reasons, this appears unfortunate nevertheless. This development is discussed in Section 2.5 just below.

### 2.5. The Current Photon Concept

The above Subsections have concentrated on the misrepresentations of the history of the early energy- and light-quantum. However, there is another issue involved which is disturbing beyond the question of historical (in-)accuracy. It is the naïve line of tradition which is typically drawn from Einstein's tentative light-quantum hypothesis in 1905 to the current understanding of "photons" as objects described by QED. In fact, many authors downright identify Einstein's light-quantum with the current photon. For example, Giancoli states with reference to Einstein's 1905 paper that "[...] this idea suggests that light is transmitted as tiny particles, or photons as they are now called" [33] (p. 775). In a similar vein Tipler and Llewellyn [14] (p. 561) remark "the Compton effect provided incontrovertible evidence for the quantization of electromagnetic radiation, the field quantum being a particle we now call the photon". Auletta et al. [28] (p. 12f) claim that the

"delocalized" classical wave is unable to account for the photoelectric effect. Emphasizing the "delocalized wave" is apparently suggesting that only a localized photon can account e.g., for the photoelectric effect.

However, as mentioned above, the early indications of light-quanta turned out to be explicable in terms of the semi-classical approximation, i.e., did not imply any quantization of the radiation field. As noted recently by Blum and Jähnert, it is curious that the very history of quantum physics was inspired by problems of radiation theory (black-body radiation, photo electricity, atomic spectra, dispersion etc.), however, the emerging theories of matrix- and wave mechanics turned out to be no theories of light and radiation—but theories of matter [57]. Within non-relativistic quantum mechanics, the photon is a foreign body anyway, and its technical foundation lies in quantum electrodynamics.

This is not the place to get into the specifics of quantum electrodynamics, but suffice it to note that there is no position operator for the photon [58] and that there is no "wavefunction" of the photon with probability interpretation in three-space [59]. The current photon cannot be localized—not even fuzzy. In addition, it is indistinguishable, while Einstein's paper from 1905 applied the statistics of distinguishable objects.

To call the photon a "particle" is rather a jargon and refers to the discreteness of the spectrum of the occupation number operator. It is rather an abstract notion which defies naïve reification. Genuine QED effects which need the field quantization (i.e., photons) for their explanation are subtle and well beyond the range of any ordinary quantum physics curriculum on high school or college level (e.g., spontaneous emission or the hyperfine splitting in hydrogen). Consequently, one is committing a technical (and not just historical) mistake if one suggests that the photoelectric effect or Compton scattering imply the need for the quantization of the electromagnetic field and that Einstein's localized and distinguishable light-quantum corresponds to what is now called photon.

This problem in the teaching of the photon concept has been long noted [60–62]. Apparently, it still prevails, and one reason for its persistence is surely the quasi-history dealt with above. There is obviously a great desire for physicists to endow their concepts with an honorable family-tree. Another famous example is provided by the concept of "atom" and "atomism". Many physicists trace the origin of modern atomism to the alleged origin in the 5th century BC, i.e., Leucippus and Democritus. However, the atoms of ancient Greece were certainly quite different from contemporary matter constituents and such a line of tradition is rather misleading.

## 3. Summary and Conclusions

At the beginning of Section 2, I sketch the common quasi-history of the early light-quantum with a four-item list of typical steps (i.e., the standard presentation of Planck's law, Einstein's light-quantum, Bohr's atomic model, and the Compton effect). Against the backdrop of my discussion, a more appropriate version of this list may sounds like this:

- In 1900, Planck introduced the energy element, $\epsilon = h\nu$, to account for the black-body spectrum. To avoid the so-called ultra-violet-he was not his intention. If Planck even intended a physical quantization is debated among historians of physics.
- Einstein's light-quantum hypothesis was not based on Planck's law but on Wien's law. His light-quanta should not be confused with the current photon concept since they were localized and distinguishable. The photo-effect did not play a prominent role in this paper either.
- Bohr's atomic model applies ideas of Einstein's theory of specific heat, but the light-quantum was rejected by Bohr until 1925. In Bohr's model, the radiation follows the frequency condition but is treated classically.
- The Compton effect (and its explanation in 1923) convinced many physicists of the reality of light quanta. With the advent of quantum mechanics in 1925/1926, the picture become more differentiated. Here, Compton effect and and photo-effect can be explained with the classical radiation field. A genuine quantum electrodynamics

(QED) effect which could be used to motivate the current photon would be, e.g., spontaneous emission.

Again, this brief account is not meant to represent the "true" history of the events. Textbooks need to simplify their subject matter, and they inevitably include inaccuracies or even errors. Additionally, in physics teaching, historical accuracy is no end in itself. In the same context, the historian of physics Helge Kragh rightly remarked [8] (p. 360):

"The purist attitude that only detailed, scholarly acceptable history should enter textbooks, amounts in practice to a denial of a historical perspective in science teaching."

However, the common quasi-history of the light-quantum in early quantum theory contains not just accidental mistakes. It systemically paints a picture of physics research as a cumulative process in which anomalies ("ultra-violet catastrophe"), successful predictions (photo-effect) or crucial experiments (Compton-scattering) drive the development inevitably towards our current understanding. In short, it is a typical Whig history. In view of current debates in physics education this may be characterized as a failure to provide an adequate view of the nature of sciense (NoS).

NoS typically refers to knowledge "about" science, as opposed to mere scientific content knowledge. Thus, NoS includes epistemological, historical and sociological aspects of science and its making. Notwithstanding the universal endorsement of NoS within the physics education literature and curriculum reform documents the very meaning of NoS remains elusive and debated. A pragmatic solution to this problem was provided by Norman Lederman and his group [63,64] who championed the so-called consensus-based NoS-view. Several authors have criticized this view and emphasized the more controversial and context dependent character of NoS; see, e.g., [65–67]. However, all approaches of NoS agree that including elements from the history and philosophy of science (HPS) can promote the teaching of NoS.

Henke and Höttecke [68] discuss major obstacles for the successful implementation of HPS elements into the classroom. One of them is a lack of adequate HPS content (e.g., case-studies or vignettes) in textbooks. Now, this case study is a reminder that there is not only a lack of adequate content but a huge problem with inadequate material. I think that the current NoS debate should pay closer attention to this issue since this problem is not confined to quantum theory.

**Funding:** This research received no external funding.

**Acknowledgments:** I thank the anonymous referees and the editorial committee for their helpful critical comments.

**Conflicts of Interest:** The author declares no conflict of interest.

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
