# Peer review of "The Quasi-History of Early Quantum Theory"

_2624-8174, doi:10.3390/physics4030057_

Round 1

Reviewer 1 Report

I read the paper with interest and joy. I do not have any critical comments. I recommend the paper, as it stands, for publication.

Author Response

Thanks for these kind remarks. I have worked on the spelling!

Reviewer 2 Report

Report on article “The Quasi-History of early Quantum Theory”

by Oliver Passon,

This work discusses early history of quantum mechanics focusing on the

history of development of the idea of quantization of electromagnetic field based 

on the works of Planck, Einstein, Bohr and Compton.  The author rightfully 

points out on discrepancies between  presentations  in several textbooks (mainly on modern physics level) 

and actual historical facts.   I found them very illuminating and useful that I can use in my 

quantum mechanics classes. 

However, I have few remarks that I hope author can take into account in some way in the revised version of  the text.  

- Criticizing  several textbooks, it gives an impression that authors of these textbooks

did poor historical research and came up with quasi-history of the events.

- However in my opinion the wast majority of textbooks use more or less standard sources of 

history of quantum theory such as George Gamow’s  “Thirty Years that Shook Physics: The Story of Quantum Theory” or “Inward Bound: Of Matter and Forces in the Physics World” of Abraham Pais.

- In this respect it is bit unclear why the author did not mention the above books at all.

-  The premise that Gamow and Pais did not know the true story of quantum theory will not be 

convincing

- Answer in my opinion lies into the concept of “History of Science” which similar to 

“History of Knowledge” or even “History of Art” should not necessarily follow closely the particular historical events. 

- Thus,  even if I completely agree  with the author that the actual time-flow of ideas are not 

the one presented in the textbooks, I will not call the currently taught  history of quantum mechanics 

a quasi-history.

Author Response

Many thanks for your comments! The question how exactly these distorted narratives originate is an intersting one indeed. However, I am not sure whether the textbook authors did any "research" in the history of physics at all. To me it seems to be the case, that the standard-narrative I presented perpetuates through the literature rather  carelessly (as discussed in Ref. 1-3). At some point it isn't questioned (and critically checked) anymore... Further more I can not support the claim that especially the books of Gamov or Pais have been used by the textbook authors. In any event: the historical remarks in textbooks are rarely backed up by any citation.   

Reviewer 3 Report

I find this article quite interesting as well as instructive.  The incisive analysis on the misrepresentations of the early stage of quantum theory found in various textbooks is illuminating and fun to read.  I thus recommend to publish this article as it is, except for one point which I hope the author reconsiders before publication.

The point concerns the characterization of the attitude of Planck toward the Rayleigh-Jeans law.  It is true that the law is officially established in 1905 by Jeans [15], but what Planck and others at the year 1900 had in mind was the Rayleigh's (incomplete) formula proposed earlier in the same year [16].   As Klein stated in [4], the energy distribution formula being proportional to the temperature T for small nu is a prerequisite for establishing the correct one which reproduces the experimental results obtained mostly by the colleagues of Planck.  Indeed, the papers of Rubens-Kurlbaum and Lummer-Pringsheim, both published slightly earlier in 1900, compare Rayleigh's formula (among other formulae) with their experimental data, which has the linear temperature behavior in line with the equipartition theorem for small nu.  

Besides, it seems that the equipartition theorem was widely accepted as a basic ingredient of statistical mechanics by 1900 already, given that it was mentioned as one of the "two clouds" in physics in the well-known lecture delivered by Kelvin.  The reason why Planck was indifferent to it appears to be attributed to his personal inclination, not because it was not widely recognized then.  As argued by Klein [4], this may underlie Planck's neglect in referring Rayleigh in his seminal papers in the year.  

Author Response

Many thanks for your effort and the kind remarks. Regarding the Rayleigh (without Jeans) law, however, I feel that my presentation is in line with both, Klein's assessment and the more recent historiography. It is certainly true that the 1900 data on bb radiation suggested lambda\propto T, as the Planck-law implies. But as Klein argues convoncingly in [4], there was in 1900 no awareness of any crisis of "classical physics" and Planck's rejection of statistical mechanics prevented him to acknowledge the point that was made by Rayleigh in his dense 1900 note. 

I also find it curious, that the referee cites Kelvin's two cloud lecture as evidence for the equipartition theorem being generally accepted. Eventually Lord Kelvin suggested in this lecture that this theorem should perhaps be abandoned! In addition the failure of the Petit-Dulong rule was an unresolved issue of that time.

Reviewer 4 Report

The paper discusses the modern misrepresentation of the initial developments of quantum theory that happened about a century ago. In my opinion, it is understandable that the narrative behind events and ideas from such a long time ago gets simplified, but it is also very important to rectify this excessive "coarse-graining" to better frame of science has actually developed and learn from it for the future. The purpose of the draft is not to rewrite history, but to give much needed insights on the background of some of the famous milestones of quantum theory. I have read the draft and enjoyed it very much, so I recommend it for publication. 

Author Response

Many thanks for your work and these kind remarks. I have worked on the spelling again.

Reviewer 5 Report

The paper concerns history of physics, not physics. Therefore for me it is not easy to evaluate the novelty of this paper. The historical facts are well known for historian (but not for average physicist) and were discussed for example in Ref. 1. On the other hand, the clear presentation of not obvious relations between Planck low of black body radiation and Einstein quantum hypothesis, or Einstein idea and Borh atomic model are interesting and can be helpful for understanding complicated way of evolution of ideas of quantum theory. So the paper is worth to publish in Physics.

Author Response

Many thanks for your kind remarks and the effort you put into it!

Round 2

Reviewer 2 Report

I recommend the article for publication. 

Reviewer 3 Report

It is true that in 1900 there was no awareness of any crisis of "classical physics" in the sense of the "UV catastrophe" and that Planck did not recognize the importance of the equipartition theorem, at least in his papers at that time, when he derived his distribution formula.  However, this is consistent with the fact that he obtained the formula so that it agreed with the experimental results in being proportional to T in the limit of small nu (Klein [4], page 465),  along the line of Rayleigh suggested earlier the same year.  Indeed, the proportionality was an important issue in the decisive paper of Rubens-Kurlbaum (Preuss. Akad. Wiss. (1900), 929), where Rayleigh's formula was mentioned among others.  

This is also consistent with the fact that the equipartition theorem was more or less "accepted" in the last years of 19th century, since otherwise there would have been no point for Lord Kelvin to treat its possible failure as the second "cloud" (irrespective of his personal view on its validity) in his lecture in 1900, taking up along with the first "cloud" concerning the relative motion of ether.  

My concern on the presentation of the author here is that, although it is correct in pointing out the common misleading statements on Planck's motivation for deriving his formula, it also may create another misleading impression that he had no intention to reproduce the proportionality of T in the formula by the sentences such as  "it played no role in Planck’s work" on page 3 in the manuscript.   Apparently, here we have several distinct things that invite confusion, that is, Rayleigh-Jeans law, Rayleigh's law, the equipartition theorem, and the proportionality of T.  I believe that the author can find it easy to remove any source of confusion regarding this matter from the present presentation.  

Author Response

see the attachment below...

Round 3

Reviewer 3 Report

I feel that the latest revision has rendered the description in question more balanced and informative.  I thus believe that the manuscript can now be published in the present form.